# Expression of the microRNA-200 Family, microRNA-205, and Markers of Epithelial–Mesenchymal Transition as Predictors for Endoscopic Submucosal Dissection over Esophagectomy in Esophageal Adenocarcinoma: A Single-Center Experience

**DOI:** 10.3390/cells9020486

**Published:** 2020-02-20

**Authors:** Daniel Neureiter, Christian Mayr, Paul Winkelmann, Bettina Neumayer, Eckhard Klieser, Andrej Wagner, Clemens Hufnagl, Klaus Emmanuel, Josef Holzinger, Oliver Koch, Tobias Kiesslich, Martin Varga

**Affiliations:** 1Institute of Pathology, Paracelsus Medical University/Salzburger Landeskliniken (SALK), 5020 Salzburg, Austria; paul.winkelmann@pmu.ac.at (P.W.); b.neumayer@salk.at (B.N.); e.klieser@salk.at (E.K.); 2Cancer Cluster Salzburg, 5020 Salzburg, Austria; 3Department of Internal Medicine I, Paracelsus Medical University/Salzburger Landeskliniken (SALK), 5020 Salzburg, Austria; christian.mayr@pmu.ac.at (C.M.); and.wagner@salk.at (A.W.); t.kiesslich@salk.at (T.K.); 4Institute of Physiology and Pathophysiology, Paracelsus Medical University, 5020 Salzburg, Austria; 5Institute of Neurointervention, Department of Neurology, Paracelsus Medical University/Salzburger Landeskliniken (SALK), 5020 Salzburg, Austria; cl.hufnagl@crcs.at; 6Department of Surgery, Paracelsus Medical University/Salzburger Landeskliniken (SALK), 5020 Salzburg, Austria; k.emmanuel@salk.at (K.E.); j.holzinger@salk.at (J.H.); o.koch@salk.at (O.K.); m.varga@salk.at (M.V.)

**Keywords:** endoscopic submucosal dissection, ESD, esophageal adenocarcinoma, epithelial-to-mesenchymal transition, EMT, E-cadherin, microRNA-200 family, microRNA-205

## Abstract

Endoscopic submucosal dissection (ESD) is an effective treatment of early esophageal adenocarcinomas (EACs). The decision of ESD over esophagectomy is based on clinical evaluation of tumor depth and invasion. On a molecular level, tumor invasion is strongly associated with epithelial-to-mesenchymal transition (EMT). Here, we investigated whether localized ESD-resected and surgically resected EAC samples displayed different expression profiles of EMT protein and microRNA markers and whether these different expression profiles were able to retrospectively discriminate localized and surgically resected samples. By doing this, we aimed to evaluate whether preoperative measurement of EMT marker expression might support the decision regarding ESD over surgery. The results showed that ESD-resected samples displayed an epithelial expression profile, i.e., high expression of epithelial protein markers, whereas surgically resected samples displayed high expression of mesenchymal markers. In addition, the anti-EMT microRNA-205 was significantly more expressed in ESD-resected samples, whereas we found no significant differences in the expression levels of microRNA-200 family members. Furthermore, in our retrospective approach, we have demonstrated that measurement of selected EMT markers and microRNA-205 has significant discrimination power to distinguish ESD-resected and surgically resected samples. We suggest that the assessment of EMT status of EAC samples on a molecular level may support clinical evaluation regarding the applicability of ESD.

## 1. Introduction

Esophageal cancer (EC) can be classified as esophageal adenocarcinoma (EAC) and esophageal squamous cell carcinoma based on localization and histomorphology [1]. EC is a serious malignancy with high mortality and increasing incidence [2]. Treatment strategies for EC are based on TNM-related tumor staging and include endoscopic treatment, surgery, chemotherapy, and chemoradiotherapy [3]. Prognosis for patients with EC strongly depends on the local invasion and the systemic spread of the tumor. Early detection and treatment of EC prior to local invasion significantly enhances prognosis for patients [4]. Over the last 20 years, endoscopic therapy has been established as a promising alternative to esophagectomy for early EC (pT1) and high-grade dysplasia (pTis) [4,5,6]. Endoscopic treatment of early EC comprises endoscopic mucosal resection (EMR) and endoscopic submucosal dissection (ESD). Both techniques are noninvasive and significantly enhance survival of patients with early EC without lymph node metastasis [7,8]. However, ESD possesses an important advantage over EMR as it allows en bloc resection of early EC regardless of size (EMR can remove lesions only smaller than 2 cm en bloc), thereby reducing local tumor recurrence [4,6]. In fact, ESD has been proven to be an efficient method for curative treatment of early EC [9,10].

Epithelial-to-mesenchymal transition (EMT) is a complex process in which cells lose their epithelial traits and gain mesenchymal characteristics accompanied by enhanced migratory and invasive capacity [11]. Hence, EMT is seen as a prerequisite for invasion and the formation of local and distant metastases [11]. The morphological and functional changes that cells undergo during EMT can be measured on a molecular level. Typically, EMT is characterized by loss of epithelial markers, such as E-cadherin and claudin-1, and by gain of mesenchymal markers, such as vimentin [11,12,13]. Moreover, several EMT-inducing factors (e.g., Snail1, Slug, ZEB1/2, Twist) contribute to successful onset and execution of EMT [14,15,16]. It is well known that microRNAs (miRs) also participate in the regulation of EMT [11]. The miR-200 family consists of five members (141, 200a, 200b, 200c, and 429) and is strongly involved in EMT by directly targeting and inhibiting ZEB1 and ZEB2, which themselves are negative regulators of E-cadherin [16,17,18].

EMT also significantly contributes to carcinogenesis of EC. Kestens et al. showed that in Barrett’s esophagus and esophageal adenocarcinoma, induction of EMT resulted in an invasive phenotype [19]. Moreover, expression of EMT markers was demonstrated in early EC and even in nonmalignant Barrett’s esophagus, suggesting that EMT might be an early event in EC progression [20,21]. Interestingly, several studies have connected the miR-200 family with EMT in EC [22]. Although it appears that members of the miR-200 family can act as both oncogenic and tumor suppressor miRs, there is strong evidence that progression of Barrett’s esophagus and EC via EMT is associated with reduced levels of miR-200 family members [23,24,25,26]. In this regard, Zhang et al. demonstrated that low expression of miR-200 family members correlated with lymph node metastasis and bad prognosis [24]. Besides the miR-200 family members, miR-205 is one of the best-described miR with tumor-suppressive and anti-invasive/EMT function in EC [27,28]. The clinical significance of miR-205 is underlined not only by the observation that the expression levels of miR-205 show a constant decline during EC progression but also by the fact that low levels of miR-205 are associated with poor prognosis of EC patients [22,28].

The decision on whether ESD is applicable over surgical resection (esophagectomy) is currently based on the local invasion depth of the tumor [4]. Identification of molecular markers that are able to distinguish early invasive (pT1) from deep invasive ECs in patient biopsies would allow preoperative identification of patients applicable for ESD over esophagectomy. In this single-center pilot study, we retrospectively compared the expression of EMT-related proteins E-cadherin, vimentin, claudin, ZEB1/2, and Snail/Slug as well as miR-200 family members and miR-205 of tumor samples from patients with localized early invasive EC (pT1) that underwent ESD with tumor samples from patients with regional invasive EC (>pT1, pN1) that underwent esophagectomy to identify differentially expressed factors that might serve as molecular markers regarding the decision of ESD over esophagectomy.

## 2. Materials and Methods

### 2.1. Patient Characteristics

Every 10 consecutive patients (*n* = 10) treated at the Department of Surgery (Salzburg, Austria) who underwent either endoscopic resection of very localized early invasive lesions (pT1, pN0) by ESD or of regional invasive (>pT1, pN1) tumors by surgical esophagectomy of the distal esophagus were included in the current study. All samples (ESD and surgical esophagectomy) originated from chemo-naïve tumors. Histological analysis of biopsies obtained prior to ESD or surgery proved the diagnosis of an adenocarcinoma. This study was performed according to local guidelines of the Paracelsus Medical University Salzburg/Salzburg County Hospital as well as in accordance with the Declaration of Helsinki (1964). The samples included in the current study were obtained during routine specimen archival procedures and were processed in an anonymized manner in this retrospective study. Briefly, the surgical technical procedure for endoscopic and surgical resection was as follows: (i) ESD was done under general anesthesia using a high-definition (HD) endoscope (Olympus GIF-HQ190, Olympus Austria GmbH, Vienna, Austria) with a transparent hood. The dissection was performed using a DualKnife (Olympus). The lesions were resected en bloc with a lateral margin of at least 1 cm. In cases of multifocal neoplasia, the Barrett’s mucosa was dissected en bloc circumferentially over the complete length of the Barrett’s segment technique (see Appendix A for exemplary images of ESD cases). (ii) All surgically treated patients received a transthoracic esophagectomy with en bloc two-field lymphadenectomy followed by gastric conduit. Reconstruction was performed with an intrathoracic anastomosis in circular technique. The current study was approved by the local ethics committee (Ref. 415-E/2370/5-2018, Federal Government of Salzburg).

### 2.2. Immunohistochemistry (IHC) Staining of EMT Markers in Formalin-Fixed Paraffin-Embedded (FFPE) Samples

Immunohistochemistry for EMT-related protein markers was performed on routinely archived FFPE tissue samples. In brief, 4 µm sections were mounted on glass slides, deparaffinized with graded alcohols, pretreated with low (pH 6.0) or high (pH 9.0) pH, and stained with the primary antibodies (listed in Table 1) on a BenchMark ULTRA platform (Ventana, Tucson, Arizona, USA) with the UltraView detection kit (Ventana). An amplification kit (Ventana) was selectively used for the antibodies SNAIL/SLUG, ZEB1, and ZEB2 (additionally with a blocking reaction (Ventana)). The results of IHC staining were carried out by assessing the extensity (% positive cells) and intensity of IHC staining (0–3) on three different representative microscope fields and expressed semiquantitatively using the quickscore method by multiplication of the extensity and intensity (yielding values between 0 and 300) for each field [29]. For Ki-67, only the extensity of IHC staining was evaluated. 

## 3. Expression Analysis of miR-200 Family Members and miR-205 in FFPE Samples

One to five 10 µm sections (depending on size of specimen) were cut from FFPE blocks using a rotator microtome after microdissection to remove surrounding nontumor tissue. Immediately afterward, microRNAs were isolated using the miRNeasy FFPE kit and the deparaffinization solution (Qiagen, Hilden, Germany) from these sections. Following photometric quality control (OD_260_ absorbance), microRNAs were transcribed using the miScript II RT system (Qiagen) according to the manufacturer’s instructions. Using miRNA-specific primers, the expression of miR-141, -200a, -200b, -200c, -205, and -429 (Qiagen) was determined by real-time RT-PCR on a ViiA7 thermocycler (Applied Biosystems, Life Technologies) in technical triplicates using the 384-well format and expressed by the 2^−ΔCt^ values related to the RNU6b housekeeping miRNA. The PCR reaction was conducted according to the manufacturer’s instructions in a 6 µL reaction containing 3 µL 2 × Quantitect SYBR Green master mix, 0.6 µL each of 10× miScript universal primer and 10× miScript microRNA-specific primer, 0.11 µL template cDNA, and 1.69 µL PCR-grade water. The PCR method included a single denaturation step (15′′/95 °C), followed by 40 cycles of amplification (15′′/94 °C; 30′′/55 °C, 34′′/70 °C) and a melt curve analysis. 

## 4. Statistics

The Mann–Whitney U test or the T-test was applied for calculating significant differences between unpaired groups of samples after testing for normality with Kolmogorov–Smirnov goodness of fit test. Correlation analysis was consecutively done using Pearson’s correlation or Kendall’s tau coefficient. K-means cluster analysis with consecutive ANOVA testing was performed for all variables to detect the centroids and to select the significant variables for the hierarchical cluster analysis. Additionally, binary logistic regression with forward selection (conditional and likelihood ratio) was done for identification of variables relevant for discrimination of the two sample groups. Finally, the Kaplan–Meier estimator was used to estimate the survival function, including factor variables. Due to the case number (*n* = 10 for each group), we performed sample size calculation (see http://jumbo.uni-muenster.de/fileadmin/jumbo/applets/fallz.html) as well as post-hoc power analysis (see http://www.psychologie.hhu.de/arbeitsgruppen/allgemeine-psychologie-und-arbeitspsychologie/gpower.html) to show the required case numbers and to demonstrate the statistical power of the used case number. Statistical calculations were carried out using OriginPro 2020 (OriginLab, Northampton, MA, USA) and SPSS Statistics version 24.0.0.1 (IBM, Vienna, Austria). Data were visualized using OriginPro 2020 (OriginLab) and Corel Designer 2018 version 20.1.0707 (Corel, Munich, Germany). Results were considered significant (*) or highly significant (**) at *p* < 0.05 and *p* < 0.01, respectively.

## 5. Results

### Cohort Description

The analyzed samples consisted of a cohort of 10 patients with EAC who underwent endoscopic resection between 2012 and 2016 (ESD, IDs = E1–E10, localized and early adenocarcinoma) and a cohort of 10 patients who underwent surgical esophagectomy between 2007 and 2015 (ID = R1–10, regional and invasive adenocarcinoma). The mean age of the patients at diagnosis was 64.2 ± 9.9 years and a female-to-male ratio of 10:1. Last check of clinical data was done in December 2019. Clinicopathological details for all cases are shown in Table 2 (for extended clinicopathological details, see Appendix A). Survival analysis showed significant differences (*p* = 0.02) between ESD-resected cases and surgically resected cases (Figure 1).

## 6. Expression Pattern of EMT Markers in Localized (ESD) and Regional (Surgically Resected) Invasive EACs

In a first step, we performed comprehensive IHC-based expression analysis of epithelial markers (E-cadherin, claudin-1), mesenchymal and EMT-promoting markers (vimentin, ZEB1, ZEB2, Snail/Slug), and cell-cycle-associated markers (p53 and Ki67) in all 20 EAC specimen (see Table 2).

Figure 2 shows exemplary IHC staining of epithelial and mesenchymal/EMT-promoting markers in localized early EAC and regional invasive EAC specimens. The epithelial markers E-cadherin and claudin-1 showed high absolute protein expression in early EACs with intensive plasma membrane positivity, whereas a circumferential reduction or complete loss of epithelial markers could be observed in regional invasive EAC. The mesenchymal markers vimentin and EMT promoters ZEB1 and ZEB2 as well as Snail/Slug showed a contrary pattern: expression levels of these factors were low in early EACs and relatively high in regional EACs. Interestingly, we observed a cytoplasmic to nuclear shift of Snail/Slug expression in regional EACs compared to early localized EACs.

Additionally, we semiquantitatively analyzed the protein expression of the selected epithelial, mesenchymal/EMT-promoting, and proliferation markers in all 10 ESD-resected (localized) and surgically resected (regional) EAC samples measured via IHC using the quickscore method to assess potential differential expression of these markers in the two specimen groups (Figure 3). 

Epithelial markers (E-cadherin, claudin-1) showed significantly higher expression in localized (ESD-resected) samples, whereas we found significantly higher expression of the mesenchymal marker vimentin and the EMT promoters ZEB1 and Snail/Slug (nuclear expression) in surgically resected and regional EAC samples. For E-cadherin, claudin-1, vimentin, and nuclear Snail/Slug expression, these differences occurred in both the tumor center and the tumor margin, whereas significantly higher levels in surgically resected samples were observed only at the tumor margin for ZEB1. Regarding the proliferation marker Ki-67, we measured significantly higher levels in surgically resected samples (tumor center). Of note, no differences in P53 expression were observed between ESD and surgically resected sample (not shown), with five cases showing no P53 expression (four ESD cases, one esophagectomy case).

## 7. Expression of miRNA Family Members in Localized (ESD) and Regional (Surgically Resected) Invasive EACs 

As the expression of EMT markers can be controlled by members of the miR-200 family and miR-205, we next asked whether ESD-resected and surgically resected specimen would also show a distinct expression profile for these miRs. As illustrated in Figure 4, there were no significant differences between the two sample groups regarding the expression levels of miR-141, miR-200a, miR-200b, miR-200c, and miRNA-429. However, localized EAC (ESD-resected) samples displayed significantly higher levels of miR-205 compared to regional tumor samples.

## 8. Correlation and Discriminative Power of EMT Markers and miRNAs

Using correlation analysis as well as cluster and linear regression analysis, we next assessed the discrimination power and consistency of our data to examine whether the observed differential expression profiles of EMT markers and miRs in localized and regional EAC specimens allow retrospective discrimination of tumor samples in ESD-resected and surgically resected groups.

Both correlation analysis of all 20 samples as well as partial correlation analysis (ESD-resected versus surgically resected) revealed a significant positive correlation between the two epithelial markers E-cadherin and claudin-1 (Figure 5). Likewise, the mesenchymal markers vimentin, ZEB1, and nuclear Snail/Slug showed a significant positive correlation with each other. Moreover, both epithelial markers E-cadherin and claudin-1 showed a significant negative correlation with vimentin, ZEB1, and nuclear Snail/Slug expression, respectively. This indicates that retrospective measurement of EMT markers in ESD-resected and surgically resected EAC samples results in robust discrimination between epithelial and mesenchymal (EMT, invasive) phenotypes. 

Although we found a highly significant positive correlation between the expression levels of all miR-200 family members among themselves, we found no significant correlation between expression of miR-200 family members and epithelial markers or mesenchymal markers. However, we detected a significant positive correlation between miR-205 expression and E-cadherin protein levels (tumor margin, Figure 5). Moreover, we found significant correlations regarding miR-205 expression and tumor size (−0.468; *p* = 0.037), T status (−0.392; *p* = 0.41), L status (−0.392; *p* = 0.41) and Pn status (−0.417; *p* = 0.30).

To further investigate the discriminative power of EMT protein markers and EMT-related miRs to retrospectively distinguish localized and regional tumors, we performed a K-means cluster analysis and logistic regression analysis. As summarized in Table 3, K-means cluster analysis with additional ANOVA identified IHC scores of vimentin (tumor center and tumor margin), E-cadherin (center and margin), claudin-1 (center and margin), Ki67 (center), ZEB1 (margin), and nuclear Snail/Slug (center and margin) as well as expression of miR-205 as factors with significant discrimination power. As illustrated in Figure 6A, these factors were able to perfectly retrospectively discriminate ESD and surgically resected samples in two clusters. We additionally performed logistic regression analysis and identified two significant factors for sample clustering: E-cadherin (margin) IHC score and miR205 expression. Although clustering of ESD samples and surgically resected samples using logistic regression was not perfect (R1 specimen in ESD sample cluster, Figure 6B), we observed better discriminatory power compared to K-means cluster analysis and, importantly, retrospective clustering analysis was possible with only two factors.

Our data indicate that different epithelial and mesenchymal/EMT-promoting marker expression profiles are able to retrospectively discriminate ESD-resected and surgically resected EAC cases. As shown in Figure 1, patients treated with ESD showed significantly longer survival. Hence, comparison of survival analyses might represent another way to test the discriminative power of our approach. Therefore, we used the cluster-discriminating IHC scores of the K-means cluster analysis (Table 3) for the significant epithelial and mesenchymal factors to define cut-off values for high/low expression cases. We then performed selective Kaplan–Meier survival analysis of ESD-resected versus surgically resected cases for epithelial (E-cadherin IHC score center/margin, claudin-1 IHC score center/margin, miR-205 expression) and mesenchymal (vimentin IHC score center/margin, ZEB1 IHC score center, nuclear Snail/Slug IHC score center/margin) markers. As shown in Figure 7A, epithelial markers were not able to significantly distinguish two patient groups regarding survival. However, selective survival analysis using mesenchymal markers resulted in robust detection of two patient groups (*p* = 0.08). The two patient groups identified via selective survival analysis using mesenchymal markers were (except one case) identical to the grouping of the patient cohort in ESD-resected and surgically resected cases, i.e., 9 out of 10 ESD cases were grouped in the low mesenchymal marker expression group in our selective survival analysis, whereas 10 out of 10 surgical cases (plus one ESD) were grouped in the high mesenchymal marker expression group. 

## 9. Discussion

ESD allows resection of early esophageal adenocarcinoma with low morbidity and enhanced quality of life compared to esophagectomy [7,9]. However, tumors have to meet certain criteria in order to be resectable by ESD. Tumor depth and lymphovascular invasion represent key parameters regarding patient selection for ESD over esophagectomy [4]. Low-grade intraepithelial neoplasia or high-grade neoplasia <20 mm without lymphovascular invasion, deep submucosal layer, or poor differentiation status are indications for possibility of ESD [31,32]. However, preoperative evaluation of esophageal adenocarcinomas is complicated as submucosal invasion is often difficult to determine [31]. As invasion status and tumor depth are key parameters regarding the decision of ESD over surgery, we speculated that EMT status of EAC might represent a possibility to evaluate the applicability of ESD over esophagectomy on a molecular level. Therefore, in the present study, we investigated whether differential expression of EMT markers can retrospectively distinguish ESD-resected (localized, pT1, pN0) and surgically resected (regional, >pT1, pN1) samples. This single-center study was designed to provide pilot data using chemo-naïve EAC samples (to avoid changes in miR and protein marker expression due to chemotherapeutic treatment) to evaluate whether expression analysis of protein and miR EMT markers is able to retrospectively distinguish ESD-resected and surgically resected samples. This initial evaluation may serve as a base for future (multicenter) studies using larger sample sizes as well as non-chemo-naïve samples to further evaluate the results of our study. 

We found that localized, ESD-resected samples showed a clear epithelial profile, e.g., high expression of epithelial markers E-cadherin and claudin-1 and low expression of the mesenchymal marker vimentin. Furthermore, these samples also displayed significantly lower levels of EMT-promoting factors ZEB1 and Snail/Slug. On the other hand, surgically resected samples of regional EACs with lymph node metastases displayed a more mesenchymal and invasive marker profile characterized by low expression of epithelial markers and high expression of vimentin and EMT promoters ZEB1 and Slug/Snail with a cytoplasmatic-to-nuclear shift. Interestingly, selective survival analysis using mesenchymal markers resulted in grouping of our cohort samples that was very similar to grouping in the ESD-resected and surgically resected cases (9 out of 10 ESD-resected cases were grouped together). Of note, the significant expression differences that were observed of epithelial and mesenchymal markers in the tumor centers between ESD-resected and surgically resected samples could be diagnostically used for therapeutic decision of preoperative diagnostic biopsies of EACs.

These data show that measurement of classical EMT markers might distinguish localized from regional EAC samples and are in line with recent reports that describe EMT and loss of epithelial markers as being involved in progression of esophageal carcinoma [20,33,34,35]. As EMT and tumor invasion represent key criteria regarding the decision of ESD versus surgery, EMT-related factors might represent attractive markers to support the decision of ESD versus surgery. Our data not only demonstrate different expression patterns of epithelial, mesenchymal, and EMT-promoting markers in localized ESD-resected and regional surgically resected tumor samples in general but also show that these expression patterns are able to retrospectively distinguish ESD samples and surgically resected samples, as demonstrated by cluster analysis and logistic regression analysis. 

Besides changes in general expression levels of EMT markers in localized and regional EAC samples, we also observed a significant shift from cytoplasmic expression (localized) to nuclear expression (regional) of the EMT promoters Snail/Slug. Interestingly, Jethwa et al. demonstrated that Slug was expressed in cytoplasm in cells of Barrett’s esophagus, whereas Slug was almost exclusively expressed in the nucleus in esophageal adenocarcinoma [35]. Moreover, they demonstrated that overexpression of Slug resulted in induction of an EMT phenotype in vitro, suggesting that nuclear expression of Slug might be an indicator of tumor progression in EAC [35]. Therefore, profiling of Snail/Slug expression (cytoplasmic versus nuclear) could be an indicator of EAC progression/invasion for decision-making regarding ESD or surgery. 

MicroRNAs are important regulators of gene expression. The miR-200 family as well as miR-205 are known to negatively regulate EMT via direct targeting of the EMT promoters ZEB1 and ZEB2, which themselves are repressors of E-cadherin [16,36]. Based on the current literature, the miR-200 family members can act as tumor-suppressive miRs and negative regulators of EMT and invasion in EAC [23,25]. Interestingly, the anti-EMT role of the miR-200 family was demonstrated to be E-cadherin-dependent but also independent of E-cadherin [37,38]. Because the miR-200 family members are described as potential invasion markers in esophageal carcinoma [27], we investigated whether, in addition to the protein EMT markers, ESD-resected and surgically resected tumor samples show different miR-200 family expression profiles that would allow retrospective discrimination of the two sample groups. However, we did not find any significant differences in the expression levels of all miR-200 family members in ESD samples versus surgically resected samples. On the other hand, measurement of miR-205 levels revealed significant differences in that miR-205 was significantly downregulated in regional tumor samples compared to localized ESD-resected samples, suggesting that miR-205 might represent a molecular marker to evaluate the applicability of ESD over surgery. MicroRNA-205 is described as possessing a tumor-suppressive role in EAC [22,28,39]. Moreover, several studies have described a progressive decline in miR-205 levels with progression of esophageal adenocarcinoma, e.g., high expression in Barrett’s esophagus and low expression in esophageal adenocarcinoma [39,40,41]. Because tumor invasion is a central aspect of progression of EAC, these observations are also reflected in our results as we were able to measure high levels of miR-205 in ESD-resected (localized) and low levels of miR-205 in surgically resected and regional invasive tumors. Thus, measurement of miR-205 levels may serve as a marker for esophageal carcinoma progression and invasion and as a potential marker for decision-making regarding ESD or surgery.

Although our initial findings highlight the potential of protein EMT factors and EMT-related miRs as molecular markers for therapeutic decision regarding ESD and esophagectomy, the present study has a largely descriptive and retrospective character. To substantiate our promising results, further studies and detailed experimental work are required. Furthermore, as the N-status is a key factor for applicability of ESD, the study was designed to retrospectively compare EMT markers in localized pT1a tumors with pN0 status and tumors with pN1 status (pT3 in our cohort). However, future studies should investigate whether the different EMT protein marker and miR expression profiles are also observable when comparing pT1a with pT1b and pT2, respectively. Moreover, we opted to use chemo-naïve tumor samples in our study to guarantee the specificity of our initial results. However, in recent years, there has been growing evidence that preoperative chemotherapeutic treatment of deep invasive EAC results in a significant clinical benefit [42]. The sample size (surgically resected samples) of our study is naturally small, and further prospective studies with neoadjuvant-treated EAC cases should be performed to support our initial results. Nevertheless, the sample size calculation (required mean case number of 13.8) and post-hoc power analysis (mean power of 82.6) that we performed demonstrate that the sample size used is statistically sufficient for the initial data that has been presented.

In conclusion, in this pilot study, we have shown that measurement of EMT markers (E-cadherin, vimentin, claudin-1, and ZEB1) as well tumor-suppressive miR-205 allows for robust retrospective discrimination of localized ESD-resected and regional surgically resected samples. Although this study has retrospective character and the sample size is small (but well defined), we suggest that preoperative assessment of expression levels of EMT-related factors and miRs might serve as potential molecular markers to support histological evaluation of tumor samples regarding the decision of ESD versus surgical resection. However, further studies with higher sample numbers and detailed experimental work are needed to substantiate our initial observations.

## Figures and Tables

**Figure 1 cells-09-00486-f001:**
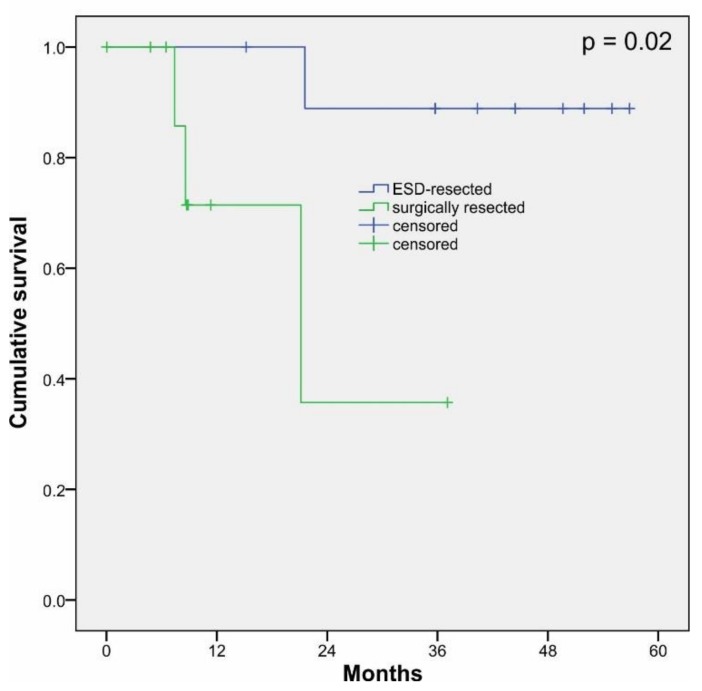
Kaplan–Meier survival analysis of endoscopic submucosal dissection (ESD)-resected cases (blue) and surgically resected cases (green) of esophageal adenocarcinoma. The analyzed cohort consisted of a total of 20 patients (*n* = 20) who underwent ESD (*n* = 10) or surgical resection (*n* = 10).

**Figure 2 cells-09-00486-f002:**
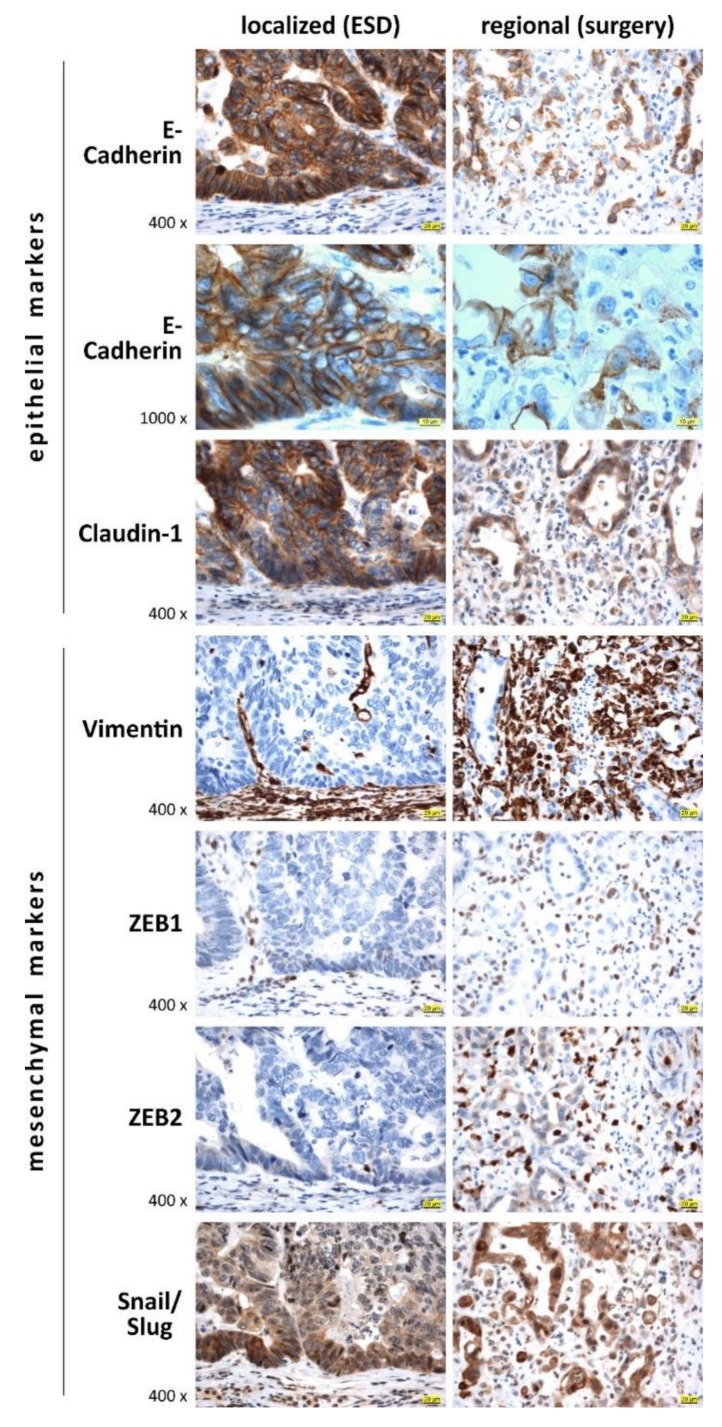
Representative IHC staining images of epithelial-to-mesenchymal transition (EMT) markers for selected localized (ESD-resected, (pT1, pN0) and regional (surgically resected (p > T1, pN1) cases of esophageal adenocarcinoma. For semiquantitative analysis of IHC staining, see Figure 3. Scale bar (yellow) indicates 20 or 10 μm for 400× and 1000× magnification, respectively.

**Figure 3 cells-09-00486-f003:**
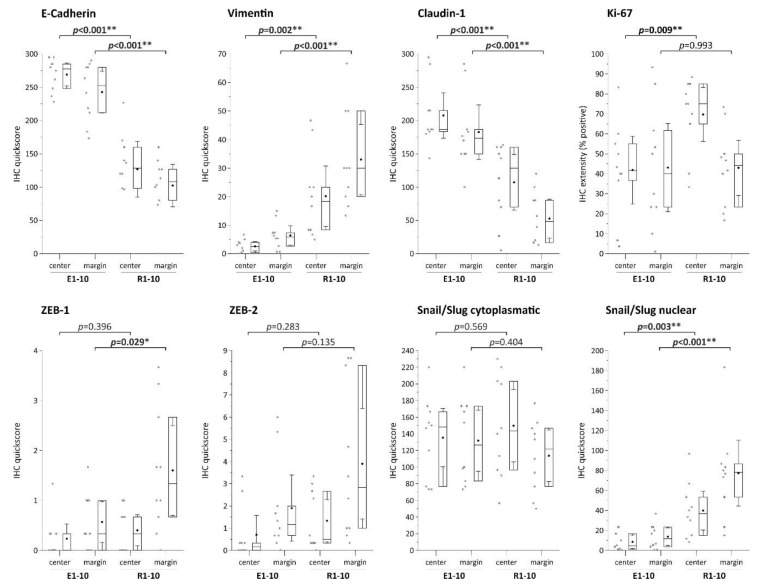
Semi-quantitative analyses of IHC staining of EMT markers using quickscore [29]. Box plots show the 25th and 75th percentiles, median (horizontal line), mean (filled circle), and 95% confidence interval (whiskers). For representative images of IHC staining, see Figure 2. * indicates significance (*p* < 0.05), ** indicates high significance (*p* < 0.01).

**Figure 4 cells-09-00486-f004:**
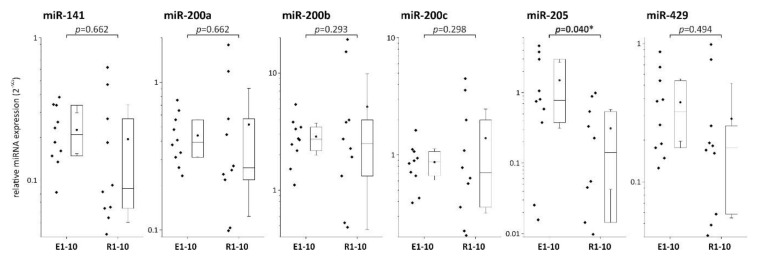
Expression analysis of microRNA (miR) family 200 and miR-205 in ESD-resected (E) and surgically resected (R) samples. * indicates significance (*p* < 0.05).

**Figure 5 cells-09-00486-f005:**
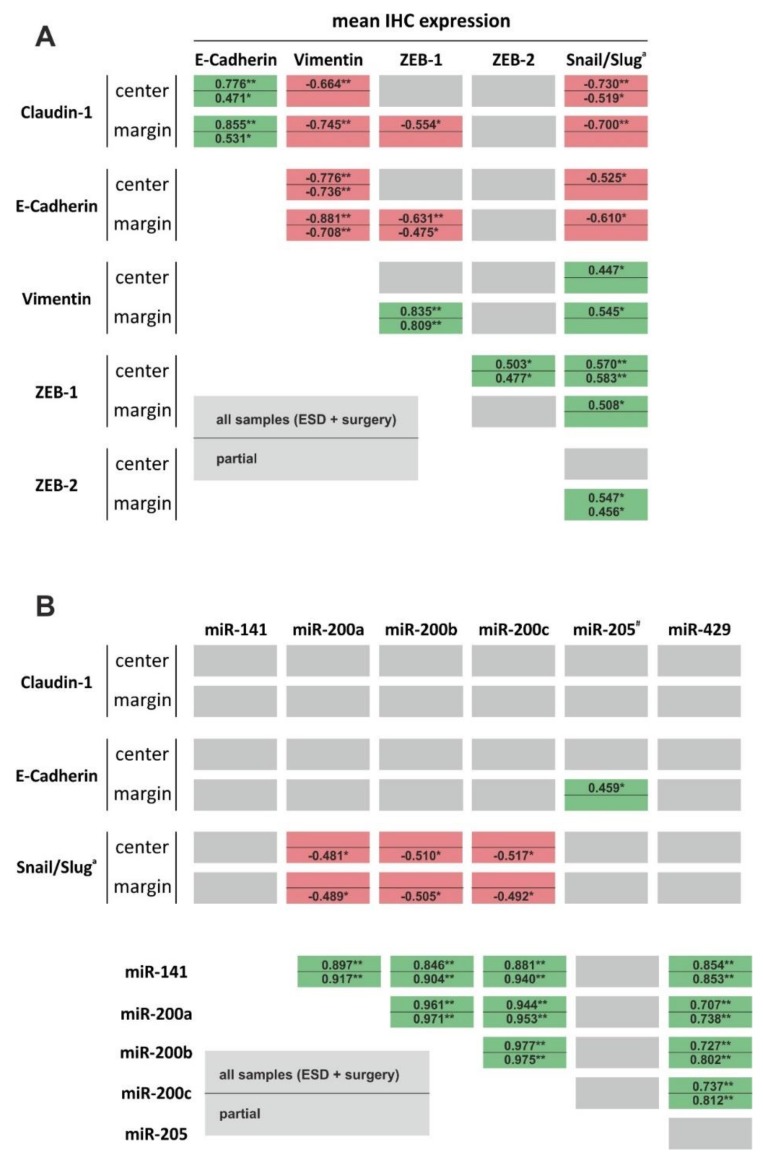
Correlation analysis between EMT markers (**A**) and EMT markers and EMT-related miRs (**B**). Green and red highlighted boxes represent positive and negative significance, respectively. Grey boxes illustrate no significant correlation. Within each box, values above the line represent correlation analysis of all samples (ESD and surgically resected samples), whereas values below the line are results of partial correlation analysis (all ESD samples versus all surgically resected samples). * indicates significance (*p* < 0.05), ** indicates high significance (*p* < 0.01). Abbreviations: EMT = epithelial-to-mesenchymal transition, ESD = endoscopic submucosal dissection, miR = microRNA. # miR with significant expression differences between ESD and surgically resected samples. ^a^ nuclear expression.

**Figure 6 cells-09-00486-f006:**
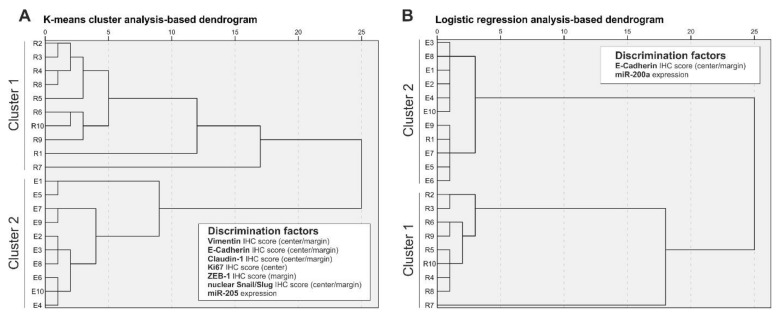
Determination of discrimination power using K-means cluster analysis (**A**) and logistic regression analysis (**B**).

**Figure 7 cells-09-00486-f007:**
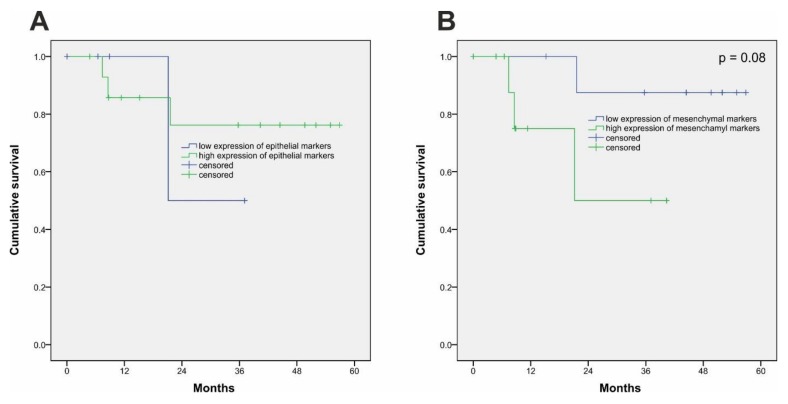
Selective survival analysis using epithelial (**A**) and mesenchymal (**B**) markers. Selection of epithelial and mesenchymal markers as well as the respective cut-off values to determine high/low cases was done based on K-means cluster analysis (Table 3). Selected epithelial markers include E-cadherin IHC score center/margin, claudin-1 IHC score center/margin, and miR-205 expression. Selected mesenchymal markers include vimentin IHC score center/margin, ZEB1 IHC score center, and nuclear Snail/Slug IHC score center/margin.

**Table 1 cells-09-00486-t001:** Immunohistochemistry (IHC) parameters.

Antibody	Vendor ^a^	Catalog No.	Species	Clone	Pretreatment	Dilution
**Claudin-1**	a	51-9000	Rabbit polyclonal	n.a.	High pH	1:200
**E-cadherin**	b	790-4497	Mouse monoclonal	36	High pH	Ready-to-use
**Ki67**	b	790-4286	Rabbit monoclonal	30-9	High pH	Ready-to-use
**P53**	b	760-2542	Mouse monoclonal	Bp53-11	High pH	Ready-to-use
**SNAIL/SLUG**	c	ab224731	Mouse monoclonal	CL3700	High pH	1:50
**Vimentin**	b	790-2917	Mouse monoclonal	V9	High pH	Ready-to-use
**ZEB1**	c	ab203829	Rabbit monoclonal	EPR17375	Low pH	1:50
**ZEB2**	c	ab230561	Rabbit monoclonal	EPR21246	Low pH	1:500

^a^ Vendors: a = ThermoFisher (Rockford, IL, USA), b = Ventana (Tucson, AZ, USA), c = Abcam (Cambridge, UK).

**Table 2 cells-09-00486-t002:** Patient characteristics with esophageal adenocarcinomas (EAC) tumor staging according to the 8^th^ edition of the TNM [30]. All cases were clinically assessed as cM0.

ID	Max. Tumor Size (mm)	Grading [G1-G3]	T	N	No. of Lymph Nodes ^a^	UICC
**endoscopic resection (ESD)**						
E1	2.4	2	1a (m2)	cN0	n.a.	IA
E2	3.7	2	1a (m2)	cN0	n.a.	IA
E3	4.3	2	1a (m3)	cN0	n.a.	IA
E4	5.1	1	1a (m2)	cN0	n.a.	IA
E5	16.2	2	1a (m3)	cN0	n.a.	IA
E6	8.1	1	1a (m2)	cN0	n.a.	IA
E7	13.2	2	1a (m3)	cN0	n.a.	IA
E8	8.3	2	1a (m2)	cN0	n.a.	IA
E9	7.2	2	1a (m3)	cN0	n.a.	IA
E10	5.1	1	1a (m1)	cN0	n.a.	IA
**esophagectomy** **(surgery)**						
R1	30	2	3	3	20/20	IVA
R2	35	3	3	1	2/21	IIIB
R3	20	2	3	2	8/55	IIIB
R4	30	3	3	3	18/24	IVA
R5	45	3	3	2	6/12	IIIB
R6	28	2	3	2	4/12	IIIB
R7	28	3	3	2	4/10	IIIB
R8	76	2	3	2	6/49	IIIB
R9	33	3	3	3	12/38	IVA
R10	50	3	3	3	22/37	IVA

^a^ The number indicates the number of lymph nodes resected during esophagectomy. Abbreviations: n.a. = not applicable, E = ESD-resected samples, R = surgically resected samples.

**Table 3 cells-09-00486-t003:** K-means cluster analysis of EMT and proliferation markers.

Factor	IHC Mean Scores		*p*
	Cluster		
	1	2	
Vimentin score (center)	20.17	2.57	0.002
Vimentin score (margin)	33.00	6.40	<0.001
E-cadherin score (center)	126.83	269.00	<0.001
E-cadherin score (margin)	102.33	242.67	<0.001
Claudin-1 (center)	107.17	207.67	0.001
Claudin-1 (margin)	52.67	182.67	<0.001
p53 (center)	170.67	122.50	0.370
p53 (margin)	158.03	131.40	0.628
Ki67 (center)	69.67	41.87	0.009
Ki67 (margin) ZEB1 (center)	43.00	43.10	0.993
	0.40	0.23	0.396
ZEB1 (margin)	1.60	0.57	0.029
ZEB2 (center)	1.33	0.70	0.283
ZEB2 (margin)	3.90	1.90	0.135
Snail/Slug cytoplasmic (center)	149.67	135.33	0.569
Snail/Slug cytoplasmic (margin)	113.67	131.83	0.404
Slug/Snail nuclear (center)	39.83	8.47	0.003
Slug/Snail nuclear (margin)	77.17	13.67	0.001
miR-141	0.194	0.226	0.662
miR-200a	0.517	0.436	0.662
miR-200b	5.154	2.869	0.293
miR-200c	1.390	0.867	0.298
miR-205	0.310	1.498	0.040
miR-429	0.285	0.375	0.494

IHC mean scores represent calculated cut-off values to distinguish the two clusters. Green and light green highlighted boxes indicate highly significant and significant discrimination factors, respectively.

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
