# Peer review of "Expression of the microRNA-200 Family, microRNA-205, and Markers of Epithelial–Mesenchymal Transition as Predictors for Endoscopic Submucosal Dissection over Esophagectomy in Esophageal Adenocarcinoma: A Single-Center Experience"

_cells, 2020, doi:10.3390/cells9020486_

Round 1

Reviewer 1 Report

Comment:

Quality of manuscript

Generally, there are too much deficit in the current papers, especially utilization of table and figure. For example, Table 2 should be summarized, and basic details should be put into supplemental data.Fig.5 should be designed and highlight the target miRNA -miR-205.

Fig.1

How many patients are included in each cohort in Fig.1?

 Fig.2

Is there any statistical data regarding to target antigen in Fig.2?

Please show scale bar in figure legend.

   4.   clinical significance of miR-205 in cancer

             Whether authors could show correlation between miR-205 and clinical parameters, such as age, gender, TNM, etc, as well as survival assay.

Author Response

We thank the reviewer for his/her comments on our manuscript. Please see below a point-to-point response.

Generally, there are too much deficit in the current papers, especially utilization of table and figure. For example, Table 2 should be summarized, and basic details should be put into supplemental data.

As suggested, we have reduced the amount of information in Table 2 and have removed L, V, Pn and R-status as well as tumor content from Table 2, and put the information in the newly created supplementary table 1.

Fig.5 should be designed and highlight the target miRNA -miR-205.

Although only miR-205 showed significant expression differences in ESD and surgically resected samples, we think that showing the correlation analysis for all tested miRs represents our initial approach (different expression patterns of miR-205 and miR-200 family) better than only showing the correlation analysis for miR-205. However, we have highlighted miR-205 in Figure 5 (both, in the figure itself and the figure description) as the significantly differently expressed miR regarding ESD versus surgically resected samples.

Fig.1:  How many patients are included in each cohort in Fig.1?

As stated in Results – Cohort description, the cohort consists of each n = 10 patients who underwent ESD and surgical resection, respectively. We apologize is this was not entirely clear and additionally added the information in the description for figure 1.

Fig.2: Is there any statistical data regarding to target antigen in Fig.2?

Please show scale bar in figure legend.

Figure 2 shows representative pictures of IHC staining that were analysed semi-quantitatively using the quickscore method (see figure 3 and text in results). To make this clearer, we have modified the figure descriptions accordingly.

We agree that the scale bar is very small and barely visible. Since we are not able to modify the size of the scale bar, we have added a description of the scale bar in the figure legend.

clinical significance of miR-205 in cancer

Whether authors could show correlation between miR-205 and clinical parameters, such as age, gender, TNM, etc, as well as survival assay.As requested, we performed additional correlation analysis for miR-205 and clinical parameters and have included the information in the result section of our manuscript.

The aim of our manuscript was a first approach to identify a set of molecular markers that support the decision-making regarding ESD versus surgical resection. Based on our results that high expression of epithelial markers robustly and retrospectively identified ESD samples and high expression of mesenchymal markers robustly and retrospectively identified surgically resected samples as well as the fact that expression differences of miR-205 were not as significant (p = 0.04) as expression differences of protein EMT markers (e.g. E-Cadherin p < 0.001) we decided not to analyse survival for individual factors including miR-205. Instead, we opted to analyse patient survival of significantly differently expressed epithelial markers and mesenchymal markers, respectively (figure 7). The expression of miR-205 as a significantly differently expressed epithelial marker is included in figure 7A.

Reviewer 2 Report

The paper deals with esophageal adenocarcinomas surgical treatment. It aims to define molecular markers that could be used in clinics to make the decision for surgeons for use of surgical procedure described as endoscopic submucosal dissection (ESD) vs esophagostomy (surgically resected esophageal adenocarcinomas). This aim is feasible, as wrong decision can lead to the spread cancer and deteriorate patients healing.

Unfortunately, this specific research (important to the close community of physicians) cannot be well accepted by authors’ claims of EMT markers based on the research of very limited sample of patients.

Thus, authors have to increase their samples to n=30 and more to make claims that EMT markers are suitable for the further validations in future publications.  I suggest focus on a few molecular markers and expand their validation on larger samples.

Minor: please describe your surgical techniques in more simple terms, as Cells readers mostly are not physicians; providing some pictures will be helpful.

Author Response

Referee 2:

The paper deals with esophageal adenocarcinomas surgical treatment. It aims to define molecular markers that could be used in clinics to make the decision for surgeons for use of surgical procedure described as endoscopic submucosal dissection (ESD) vs esophagostomy (surgically resected esophageal adenocarcinomas). This aim is feasible, as wrong decision can lead to the spread cancer and deteriorate patients healing.

We thank the reviewer for his/her comments on our manuscript. Please see below a point-to-point response.

Unfortunately, this specific research (important to the close community of physicians) cannot be well accepted by authors’ claims of EMT markers based on the research of very limited sample of patients.

Thus, authors have to increase their samples to n=30 and more to make claims that EMT markers are suitable for the further validations in future publications.  I suggest focus on a few molecular markers and expand their validation on larger samples.

We agree with the reviewer that further studies with increased sample sizes are required to substantiate our results. Our study represents a first (retrospective) approach to identify molecular markers that support the decision-making regarding ESD or surgical resection. By showing that specific EMT markers and miRNA-205 are able to robustly distinguish ESD and surgically resected samples we provide first data that should be further validated in other patient cohorts with a greater sample size. Furthermore, all patient samples (ESD and surgically resected) are chemo-naïve to guarantee the specificity of our results, which naturally results in small sample sizes (especially for surgically resected samples), as most non ESD-resectable esophageal adenocarcinomas are now treated with chemotherapy pre-operative as the standard therapeutic strategy. We have added this information in the patient characteristics section of the manuscript. Moreover, due to the time restrictions of the journal for re-submission of the manuscript (8 days) we are not able to include more patient samples in our analysis. However, we have emphasized on your point in the discussion.

Minor: please describe your surgical techniques in more simple terms, as Cells readers mostly are not physicians; providing some pictures will be helpful.

As requested, we have modified the methodical section of the manuscript regarding the surgical techniques and provided a more simple description of the surgical procedures. In addition, we have provided exemplary images of two ESD cases (E9 and E10) before and after ESD as well as of the corresponding ESD resectated (see supplementary figure 1).

Reviewer 3 Report

Manuscript ID: cells-679848
Expression of the microRNA-200-family, microRNA-205 and markers of epithelial-mesenchymal transition as predictors for endoscopic submucosal dissection over esophagectomy in esophageal adenocarcinoma

Interesting studies in which the Authors investigated whether ESD-resected and surgically resected EAC samples collected in recent years display different expression profiles of EMT protein and micro-RNA markers and whether these different expression profiles are able to retrospectively discriminate localized and surgically resected samples to evaluate whether pre-operative measurement of EMT marker expression might support the decision regarding ESD over surgery.

Questions:
Materials and Methods-miRNA analysis: what amount of sample was taken for RNA isolation; what reagents were used for the PCR reaction; describe briefly qPCR procedure. Have isolated RNA samples been validated for quality?

Some R group samples were 5 years older than the E group samples. Could this affect the results (RNA degradation)?

Figure 2 Scale not visible in the pictures.

Author Response

Referee 3:

Expression of the microRNA-200-family, microRNA-205 and markers of epithelial-mesenchymal transition as predictors for endoscopic submucosal dissection over esophagectomy in esophageal adenocarcinoma

Interesting studies in which the Authors investigated whether ESD-resected and surgically resected EAC samples collected in recent years display different expression profiles of EMT protein and micro-RNA markers and whether these different expression profiles are able to retrospectively discriminate localized and surgically resected samples to evaluate whether pre-operative measurement of EMT marker expression might support the decision regarding ESD over surgery.

We thank the reviewer for his/her comments on our manuscript. Please see below a point-to-point response.

Questions:

Materials and Methods-miRNA analysis: what amount of sample was taken for RNA isolation; what reagents were used for the PCR reaction; describe briefly qPCR procedure. Have isolated RNA samples been validated for quality?

We have modified the materials and methods analysis section as requested. The amount of FFPE sample for miR analysis is described in the manuscript (i.e. one to five 10 µm sections including mainly tumor tissue, depending on the size of the specimen; see Material and Methods section). Isolated samples have been validated for quality via photometry showing no obvious differences between R and E sample groups. Details on the qPCR procedure are now provided in the manuscript.

Some R group samples were 5 years older than the E group samples. Could this affect the results (RNA degradation)?

Results from the photometric sample quality control as well as analysis of the expression levels of the housekeeping miR indicated now obvious differences between target concentrations of R and E samples. Therefore, we suggest that the different age of the FFPE sample did not influence the quality and quantity of target miRs in the samples.

Figure 2 Scale not visible in the pictures.

We agree that the scale bar is very small and barely visible. Since we are not able to modify the size of the scale bar, we have added a description of the scale bar in the figure legend.

Round 2

Reviewer 1 Report

Authors answered all questions.

Author Response

Authors answered all questions.

We thank the reviewer for his/her positive comment. No further changes required.

Reviewer 2 Report

Authors have improved minor points of my concern of their manuscript; however, they did not provided larger sample analysis.

This is serous flaw and the manuscript cannot be considered for the publication before the addressing this issue.

Authors should take more time to improve their interesting work.
